# First Detection of Hepatitis E Virus (*Rocahepevirus ratti* Genotype C1) in Synanthropic Norway Rats (*Rattus norvegicus*) in Romania

**DOI:** 10.3390/v15061337

**Published:** 2023-06-07

**Authors:** Daniela Porea, Cristian Raileanu, Luciana Alexandra Crivei, Vasilica Gotu, Gheorghe Savuta, Nicole Pavio

**Affiliations:** 1Department of Public Health, Faculty of Veterinary Medicine, Iasi University of Life Sciences, 700490 Iași, Romania; daniela.porea@yahoo.com (D.P.); cristian.raileanu@yahoo.com (C.R.); criveilucianaalexandra@gmail.com (L.A.C.); epirovet@yahoo.com (G.S.); 2Laboratories and Research Stations Department, Danube Delta National Institute for Research and Development, 820112 Tulcea, Romania; 3Regional Center of Advanced Research for Emerging Diseases, Zoonoses and Food Safety Iași, University of Life Sciences, 700490 Iași, Romania; 4Department of Parasitology and Parasitic Diseases, Faculty of Veterinary Medicine, University of Agronomical Sciences and Veterinary Medicine of Bucharest, 011464 Bucharest, Romania; gotuvasilica@yahoo.com; 5Agence Nationale de Sécurité Sanitaire de L’alimentation de L’environnement et du Travail (ANSES), Institut National de Recherche pour L’agriculture L’alimentation et L’environnement (INRAE), École Nationale Vétérinaire d’Alfort (ENVA), UMR Virology, 94700 Maisons-Alfort, France

**Keywords:** hepatitis E virus, rats, *Rocahepevirus*, phylogenetic analysis, Romania

## Abstract

Hepatitis E virus (HEV) is an emerging zoonotic pathogen with different viral genera and species reported in a wide range of animals. Rodents, particularly rats, carry the specific genus rat HEV (*Rocahepevirus genus,* genotype C1) and are exposed occasionally to HEV-3 (*Paslahepevirus genus*, genotype 3), a zoonotic genotype identified in humans and widely distributed in domestic and feral pigs. In this study, the presence of HEV was investigated in synanthropic Norway rats from Eastern Romania, in areas where the presence of HEV-3 was previously reported in pigs, wild boars and humans. Using methods capable of detecting different HEV species, the presence of HEV RNA was investigated in 69 liver samples collected from 52 rats and other animal species. Nine rat liver samples were identified as being positive for rat HEV RNA (17.3%). High sequence identity (85–89% nt) was found with other European *Rocahepevirus*. All samples tested from other animal species, within the same environment, were negative for HEV. This is the first study to demonstrate the presence of HEV in rats from Romania. Since rat HEV has been reported to cause zoonotic infections in humans, this finding supports the need to extend the diagnosis of *Rocahepevirus* in humans with suspicion of hepatitis.

## 1. Introduction

Hepatitis E virus (HEV), the causative agent of hepatitis E in humans, is a single-stranded positive sense RNA virus that belongs to the *Hepeviridae* family. HEV genome comprises three open reading frames (ORFs), encoding the non-structural polyprotein (ORF1), the capsid protein (ORF2) and the phosphoprotein (ORF3). According to the current data of the International Committee on the Taxonomy of Viruses (ICTV), the family *Hepeviridae* includes two subfamilies, *Orthohepevirinae* and *Parahepevirinae*, and five genera, *Paslahepevirus*, *Avihepevirus*, *Rocahepevirus*, *Chirohepevirus* and *Piscihepevirus* [1]. Most of the HEV strains identified belong to the *Paslahepevirus* genus (previously named *Orthohepevirus A*), which is divided into two species, *P. balayani* and *P. alci*, and have been isolated mainly from humans, domestic pigs, wild boars, rabbits, deer, and camels. The *P. balayani* species comprises eight different genotypes (HEV-1 to HEV-8), of which four main genotypes (HEV-1 to HEV 4) can infect humans [2]. The first two genotypes (HEV-1 and HEV-2) are restricted to humans and are endemic to developing countries, while the other two genotypes (HEV-3 and HEV-4) are zoonotic, infect humans, domestic pigs, wild boar, deer and rabbits, and cause sporadic cases worldwide [3]. Genotypes HEV-5 and HEV-6 infect wild boars, HEV-7 infects dromedary camels and humans, while HEV-8 infects Bactrian camels and cynomolgus macaques [4]. The species *P. alci* has been proposed for an HEV strain isolated from moose in Sweden. The other three genera of the subfamily *Orthohepevirinae* infect birds (*Avihepevirus*), rodents, shrews and carnivores (*Rocahepevirus*) and bats (*Chirohepevirus*) [1]. Of these, the *Rocahepevirus* genus is not restricted to its main hosts, but the occurrence of strains of this genus in humans has been reported as well [5]. Members of the subfamily *Parahepevirinae* infect trout and salmon [1].

The *Rocahepevirus* genus (previously named *Orthohepevirus C*, HEV-C) is divided into two species, *R. eothenomi* and *R. ratti*, and the *R. ratti* species comprises two genotypes (HEV-C1 and -C2) [6]. HEV-C1 has been detected in rodents (*Rattus* sp., *Bandicota indica*), eulipotyphlids (musk shrew, *Suncus murinus*) and humans, while the C2 genotype has been detected in mustelids (ferret and mink). HEV-C2 has been detected in companion ferrets (*Mustela putorius*) in the Netherlands, the USA and Japan and in farmed minks, but not in wild ones, in Denmark [7,8]. Strains related to ferrets and rat hepeviruses have also been detected in foxes in The Netherlands and Germany [9,10]. The specie *R. eothenomi* has been detected in various vole species (e.g., *Microtus arvalis*, *Eothenomys melanogaster* and *Eothenomys inez*) in Hungary, Germany, the Czech Republic and China, and in Chevrier’s field mouse (*Apodemus chevrieri*) [11,12,13]. Two putative genotypes, HEV-C3 and HEV-C4, can be defined within this specie [12].

The main host of the *R. ratti* species, HEV-C1, is rodents, and although it has been identified in humans, the zoonotic potential of rat HEV strains remains to be better investigated [5,14]. The detection of rat HEV in humans with acute hepatitis and the worldwide distribution of rat HEV in rodents suggest that rat HEV infection may be an emerging disease [5]. Infections with rat HEV in humans have been described in Hong Kong, Canada, Spain and France in both immunocompetent and immunocompromised patients [5,15,16,17]. Severe outcomes were described, with fatalities in at least two cases [5,18] and chronic infection in one case [17].

The first recording of HEV in rodents was documented in Germany in 2010, with the identified strains belonging to rat HEV [19]. In the Norway rat (*Rattus norvegicus*), the main rodent carrying HEV, both rat HEV and HEV-3 were detected [4]. Since then, rat HEV has been detected in several countries, and in different species of rodents: in Norway rats and black rats (*R. rattus*) from Germany and other European countries (France, Denmark, Austria, Switzerland, the Czech Republic, Belgium, Greece, Lithuania, Great Britain, Italy and Hungary), the USA and Indonesia, tanezumi rats (*Rattus tanezumi*) from Vietnam, and bandicoot rats (*Bandicota indica*), *Rattus flavipectus*, and *Rattus rattoides losea* from China [20,21,22,23,24,25,26]. At the same time, a series of studies have reported the presence of HEV-3 in rodents in different parts of the world: in Norway rats from Japan, the USA, Belgium and Great Britain, in black rats from the USA and Italy, in house mice (*Mus musculus*) from Great Britain, in yellow-necked mice *(Apodemus flavicollis)* from Croatia and in capybaras (*Hydrochoeris hydrochaeris*) from Brazil [22,24,25,27,28,29,30]. Some of these studies, based on the detection of HEV-3 only in the intestine/feces but not in the liver, suggested that the presence of HEV-3 in rodents was associated with the ingestion of swine feces containing the virus rather than with infection [24,25].

In addition, serological studies have been conducted to determine HEV seroprevalence in various species of Rodentia (*Muridae* and *Cricetidae*) and Soricomorpha. According to the data summarized by Wang et al., until 2020, 23 species from eight countries (the USA, Japan, Germany, China, Vietnam, Lithuania, Indonesia, and India) were found to be positive for anti-HEV antibodies. The most investigated species were *Rattus norvegicus* and *Rattus rattus*. These two species present the highest seroprevalence rates recorded (in the USA, up to 77% and 90%, respectively) [6].

The role of rats in the transmission of the zoonotic HEV-3 genotype, the most commonly detected genotype in humans worldwide, the main animal reservoir of which is domestic pigs [14,31], also needs to be clarified.

In Romania, the HEV has been recorded, following the investigation and identification of the HEV-3 in pigs, wild boars and humans, especially in the Eastern region of the country [32,33,34]. Based on these records, we pointed out that pigs and wild boars could be an endemic source of human HEV-3 infections [33]. With this new study, we aimed to identify and characterize the circulating HEV strains in rats, in order to assess the role of this species in the epidemiology of HEV transmission to humans or pigs in the studied area. Moreover, this study aimed at the identification of the HEV in other wild animals living in the same habitats as rats.

## 2. Materials and Methods

### 2.1. Sample Collection

In 2016, a total of 69 liver samples were collected from seven wild animal species in different counties in Eastern Romania, as follows:-Fifty-two rats—Norway rats (*Rattus norvegicus*) from four counties: Suceava, Iasi, Bacau and Tulcea (Figure 1, Table 1);-Nine foxes (*Vulpes vulpes*) from Vrancea county (Figure 1);-Four ferrets (*Mustela putorius*), one from Iasi County and three from Tulcea County, one coypu (*Myocastor coypus*) and one muskrat (*Ondatra zibethicus*) from Tulcea County, one mole (*Talpa europaea*) from Suceava County and one mouse (*Mus musculus*) from Iasi County (Figure 1).

**Figure 1 viruses-15-01337-f001:**
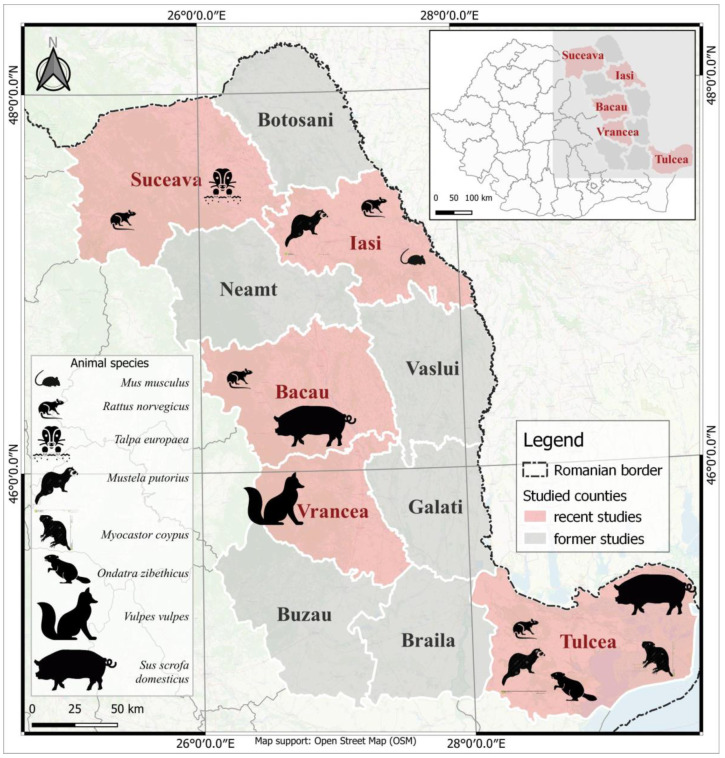
Geographical distribution of the counties included in the study, with the representation of the analyzed species. This figure was created using the Quantum Geographic Information System (GIS) (version 3.28.5-Firenze). (Animal pictograms sources: www.flaticon.com, www.thenounproject.com/icon/ and www.dreamstime.com/, accessed date: 28 March 2023).

**Table 1 viruses-15-01337-t001:** Data on the tested samples from rats.

No.	Location/Site	Number of Rats/Location/Counties			
		Suceava	Iasi	Bacau	Tulcea
1	Intensive system pig farm	-	-	-	12
2	Semi-intensive system pig farm	5	-	11	-
3	Extensive system pig farm	1	-	5	-
4	Farm with different animal species, forestry site	10	-	-	-
5	Cattle farm, metropolitan area	-	1	-	-
6	Urban settlements, near garbage cans	-	5	-	-
7	Rural settlements, population’s household	-	-	-	2
**Total**		**16**	**6**	**16**	**14**

The rats were trapped in the course of pest control programs, the foxes were legally hunted and collected by the National Sanitary Veterinary Authority during the rabies monitoring program, and the other species were either road-killed or died a natural death. Necropsy and collection of tissue samples from rodents were performed in the Department of Public Health of the Faculty of Veterinary Medicine, Iasi University of Life Sciences. The animal species were identified based on their morphological characteristics. Rats were trapped at various locations near human settlements and animal farms (Table 1).

In 2017, 10 pool fecal samples were collected from two indoor pig farms (*Sus scrofa domesticus*) where rats were trapped, six from a semi-intensive pig farm in Bacau County and four from an intensive pig farm in Tulcea County. The samples collected from Bacau came from fattening pigs, and those from Tulcea came from growing and fattening pigs. Each pool contained samples from 10 individuals.

Following collection, the samples were stored at −80 °C until further testing.

### 2.2. RNA Extraction and Gene Amplification—PCR Assays

RNA extraction from liver samples followed the protocol previously described by Porea et al., 2018 [33]. For fecal samples, the protocol previously described by Aniţă et al., 2014 was used [32]. Total RNA was extracted from 30 mg liver samples or 200 μL 10% fecal suspensions using the RNeasy Mini kit (Qiagen, Courtaboeuf, France) and QIAamp Viral RNA Mini kit (Qiagen), respectively. RNA was eluted twice with 40 μL of RNase-free water after 1 min of incubation at room temperature. RNA was aliquoted and stored at −80 °C until use.

For the molecular detection of HEV RNA from liver samples, three different assays were used. A TaqMan real-time RT-PCR assay [35,36] and a RT-nested PCR [37,38] specific for HEV genotypes 1–4 were performed as previously described by Porea et al., 2018 [33]. The RT-nested PCR assay was also used to detect HEV RNA from pig faecal samples. The TaqMan real-time RT-PCR assay targets a fragment of ORF3 and the RT-nested PCR one of the ORF2, which are highly conserved regions in different HEV genotypes (Appendix A).

A nested broad-spectrum RT-PCR (NBS-RT-PCR) for hepeviruses, including rat HEV, was also performed by amplifying a fragment of ORF1 with two sets of degenerate primers [39] (Appendix A). The assay was adapted from the method previously described by Johne et al., 2010 [39]. The cDNA was synthesized according to the method previously described by Barnaud et al. 2019 [36]. The first round of PCR was performed with the outer pair of primers (HEV-cs and HEV-cas) using the Platinum^®^ Taq DNA Polymerase kit (Life Technologies, Villebon-sur-Yvette, France). The thermal profile included 95 °C for 5 min, followed by 40 cycles of 94 °C for 30 s, 50 °C for 30 s and 74 °C for 45 s, with a final incubation at 74 °C for 5 min. The second round of PCR was performed with the inner primer pair (HEV-csn and HEV-casn) by using the thermal profile previously described by Johne et al., 2010 [39].

Each experiment included several control samples. For the detection of HEV in wild animals, negative and positive wild boar samples, stored at the UMR Virology, Maisons-Alfort, France, were used [40]. For the detection of HEV in pigs, a sample (roFPR4) stored in the molecular biology laboratory of the Faculty of Veterinary Medicine Iaşi, Romania was used as a positive control [32].

The PCR products were analyzed after migration on agarose gel and ethidium bromide staining (Appendix A).

### 2.3. Nucleotide Sequencing and Phylogenetic Analysis

Sequencing of the amplicons with the expected size was performed by MWG Biotech AG (Eurofins MWG, Ebersberg, Germany) using the Sanger method. Sequencing of PCR products was performed for each direction. Specific inner primer pairs were used for the sequencing of positive RT-nested PCR amplicons. Nucleotide sequences were analyzed and individually edited using Bioedit software (version 7.2.5). Sequences were uploaded into the alignment explorer of MEGA 7 [41] and aligned with MUSCLE [42]. A phylogenetic tree was built using the neighbor-joining method based on the Tamura Nei model [43] with 1000 replicates. Reference sequences for members of the subfamily *Orthohepevirinae* were used [1].

## 3. Results

### 3.1. Detection of HEV RNA

The two assays specific for HEV genotypes 1–4 revealed that all liver samples were negative for these genotypes. Additionally, the RT-nested PCR assay revealed that the fecal samples from the pigs were also negative.

Using the nested broad-spectrum RT-PCR assay, HEV RNA was detected only in liver samples from rats (9/52, 17.30%). Positive rats originated from each analyzed county and from four of the seven sites studied that were located near livestock farms (Table 2).

### 3.2. HEV Sequence Analysis

The nine RT-nested PCR products that were positive for HEV RNA were sequenced. BLAST analysis showed that all isolates belonged to the rat HEV-C1. Nucleotide identity between our isolates and homologs in the GenBank database ranged from 80% to 89%. This was observed with respect to human isolates from Spain (80%) and to rat isolates from Denmark, Germany, Belgium, The Netherlands, the USA and China (85–89%) (Figure 2 and Table 3).

The obtained isolates were clustered according to county (Figure 2) with nucleotide identities ranging from 85 to 91%. Interestingly, the highest nucleotide identity (91%) was found between the isolate from a rat in Iasi County (SisD, GenBank accession number OQ601523), captured near a cattle farm, and the isolates from rats in Tulcea County (StlF-1, StlF-5 and StlF-7, GenBank accession numbers: OQ601529, OQ601530, and OQ601531, respectively) caught near an intensive pig farm (Figure 2 and Table 3). These isolates had a maximum nucleotide identity of 89% with an isolate obtained from a rat in Denmark (GenBank accession number KC294199.1). Additionally, the isolate from a rat in Iasi County had 87% nucleotide identity with two isolates from rats in the United States (GenBank accession numbers JF516246 and KM516906.1) and one isolate from rat in China (GenBank accession number MH729811).

The isolates obtained from rats in Suceava County (Ssv4-1, Ssv4-2 and Ssv4-8, GenBank accession numbers: OQ601526, OQ601527, and OQ601528, respectively) had a maximum nucleotide identity of 89% with two isolates obtained from rats in China and the Netherlands (GenBank accession numbers OM037396 and ON644869) (Table 3).

Isolates obtained from rats in Bacau County (SbcS-1 and SbcS-6, GenBank accession numbers OQ601524 and OQ601525) had a maximum nucleotide identity of 87% with two isolates obtained from rats in Germany and Belgium (GenBank accession numbers KX774658, and KY938022, respectively) (Table 3).

## 4. Discussion

In the epidemiology of HEV infection, the role of rodents is currently debated worldwide, as the presence of the HEV-3 genotype in rats as well as the rat HEV genotype in humans has been described.

In this study, we sought to identify the HEV strains circulating in rodents from Eastern Romania, where we previously reported the presence of HEV-3 in pigs, wild boars, and humans [32,33,34]. Among the analyzed species, HEV RNA was detected only in rat liver samples. Sequencing of the obtained amplicons highlighted that all isolates belonged to the rat HEV-C1 genotype, and no strains belonging to other genotypes were identified. Previous study conducted by Aniţă et al., 2014 [32] in Romania showed that HEV strains identified in pigs belong to HEV-3, and several situations have been described worldwide in which HEV-3 was found in rats living near animal farms or human. The study published by Kanai et al., 2012 [27] showed that strains isolated from rats captured near a pig farm in Japan were genetically related to the pig strain, which belongs to HEV-3. On the other hand, the study by Li et al. 2013 [44] in China found that HEV strains identified in rats caught near both garbage cans and pig farms were not related to HEV strains identified in other mammalian species. The detection of an HEV-3 sequence in a sample taken from a rat in Belgium highlighted the need for further studies to assess the possibility of “spillover” infection of rats with human pathogenic hepeviruses and its public health implications [22]. Furthermore, De Sabato et al., 2020 [25] found that the intestinal contents of two rats were positive for rat HEV and HEV-3 RNAs, while the liver samples of both animals were negative. Thus, rodents could be exposed to and infected with HEV-3 but are not responsible for sufficient viral amplification and a source of transmission to other hosts. In our study, similar to the cited studies, we focused on assessing the circulation of rat HEV and HEV-3 in rats rather than determining rat HEV in swine and other animals. Indeed, the presence of the virus has been investigated in humans in a limited number of countries, including Hong Kong, France, Spain and Germany, and has been detected in patients with acute hepatitis in Hong Kong and Spain and in chronic hepatitis in France [5,15,17,45,46]. The transmission pathway of rat HEV from natural host to human remains to be identified.

The phylogenetic and blast analyses revealed that the nucleotide identity between our isolates and the homologues in the GenBank database ranged from 80% to 89% and that our isolates were related to rat HEV strains circulating in Europe. The nucleotide identity between our isolates ranged from 85% to 91%, and phylogenetic analysis showed that they were clustered by county (Figure 2). In Germany, phylogenetic analysis of partial sequences and the whole rat HEV genome showed clear clustering of isolated strains according to the geographic sampling area [19,47]. In France, the increased proportion of homology between isolated strains was correlated with the occurrence of HEV in rats captured at a single site [21]. Isolated strains from rats in France are genetically related to rat isolates from Germany and Denmark and are divergent from a strain identified in Vietnam [21]. Phylogenetic analysis showed that these observations are also similar to our isolates (Figure 2). In general, genome diversity of rat HEV has been observed in several studies, and phylogenetic analysis revealed three distinct clusters within this genotype C1, which could constitute three possible subtypes (GI, GII, GIII) [14,44,48]. The diversity between genotypes is 19.5–23.5 (22.0 ± 1.7)% [48]. Phylogenetic analysis showed that the sequences from Romania are closely related to rat HEV sequences from Northern Europe, such as from Denmark [49] or Belgium and Germany (Figure 2, Table 3). These sequences clustered within the proposed subtype GI, but are more distant from those of Southern Europe such as Spain and may constitute a new European clade or subtype. The analysis of complete genomes would improve this comparison. As rat populations may have followed trade routes, the identification of other sequences from different countries would contribute to drawing a possible map of rat HEV virus evolution and spread.

In this study, HEV RNA was detected in 9 of the 52 samples collected from rats. The prevalence (17.30%) is generally consistent with prevalence rates previously reported in European countries [22]. The results showed that HEV was present in rats from all of the studied counties (Table 2), indicating a broad geographical distribution of rat HEV in this species in the study area. The presence of HEV RNA in rats captured from different locations (Table 2), regardless of the number of animals studied, suggests that the virus circulates in these rodent species with various habitats. Similar observations were made in studies in Germany, where HEV was detected in rats caught in all cities studied in northeastern and southwestern Germany [47]. The literature indicates that the virus has been detected in rats caught near urban and rural settlements in Germany and other European countries [22], near pig farms in Japan [27], and in the proximity of urban and rural settlements and waste and pig farms in China [44]. Similar to recent studies in Italy [25], Lithuania [23] and Great Britain [24], in our study, rat HEV was identified in rats captured near animal farms.

Samples from the other species, included in this study, were identified as HEV RNA-negative. For three of the species, HEV was previously reported in other European countries, namely HEV-C2 in ferrets (*Mustela putorius*) in the Netherlands [7], HEV-3 in house mice (*Mus musculus*) in the UK [50], and strains related to ferrets and rat HEV in foxes (*Vulpes vulpes*) in the Netherlands and Germany [9,10]. In two other studies, despite previous reports, HEV was not detected in house mice in the UK [24] and rats in Austria [51], and factors such as variability in assay sensitivity and investigation of a single tissue type were suggested. Additionally, the results of the study conducted by Lhomme et al. in 2015 [52] showed that HEV RNA was not detected in the 78 samples collected from French coypu. The authors of the study suggested that one of the possible reasons for this result could be the non-specificity of the primers used to detect HEV in these rodents. In the present study, the small number of samples, by animal species, is a limitation. It would be recommended to pursue this work with a larger collection of samples.

## 5. Conclusions

The first detection and molecular characterization of rat HEV in Romania represent the novelty of the present study.

The results of the molecular investigations regarding the presence of hepatitis E virus in rats allowed the classification of the isolated strains into the rat HEV genotype and indicated the presence of a new source of infection in the study area.

To better assess the role of rats in the epidemiology of HEV infection in humans, it is recommended to screen for rat HEV in humans with a suspicion of hepatitis and other animals in close contact with rats.

## Figures and Tables

**Figure 2 viruses-15-01337-f002:**
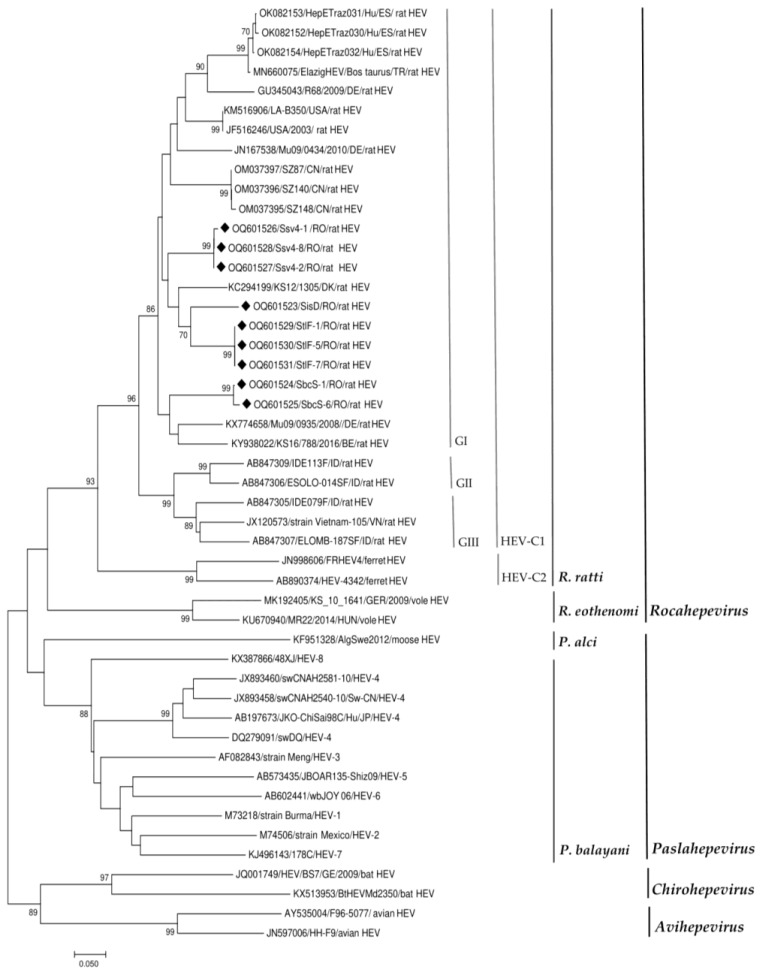
Phylogenetic tree based on the alignment of the generated sequences with the best hits to our sequences in GenBank and with reference sequences for the members of the subfamily *Othohepevirinae* [1]. Countries of origin of the rat HEV sequences used are as follows: RO, Romania; DE, Germany; BE, Belgium; DK, Denmark; ES, Spain; TR, Turkey; USA, the United States; CN, China; ID, Indonesia; VN, Vietnam (the rat HEV strains obtained in our study are highlighted by black diamonds). Putative rat HEV-C1 genotypes are indicated GI, genotype 1; GII, genotype 2; GIII, genotype 3. The phylogenetic tree was built using the neighbor-joining method based on the Tamura Nei model with 1000 replicates. The tree is drawn to scale, with branch lengths in the same units as those of the evolutionary distances used to infer the phylogenetic tree.

**Table 2 viruses-15-01337-t002:** Molecular detection of HEV genome.

Species	Type of Samples	County				
**Wild**	Liver	Suceava	Iasi	Bacau	Tulcea	Vrancea
** *Rattus norvegicus* **		3/16 (3/10 ^4^)	1/6 (1/1 ^5^)	2/16 (2/11 ^2^)	3/14 (3/12 ^1^)	-
** *Vulpes vulpes* **		-	-	-	-	0/9
** *Mustela putorius* **		-	0/1	-	0/3	-
** *Mus musculus* **		-	0/1	-	-	-
** *Talpa europaea* **		0/1	-	-	-	-
** *Myocastor coypus* **		-	-	-	0/1	-
** *Ondatra zibethicus* **		-	-	-	0/1	-
**Domestic**						
** *Sus scrofa domesticus* **	Pool faecal	-	-	-	0/10	-

^1, 2, 4, 5^ Numbers corresponding to the different settings indicated in Table 1 (column 1).

**Table 3 viruses-15-01337-t003:** HEV sequence BLAST analysis.

Sample Number	County	Accession Number	Best Sequence Identity (%) Country
SiD	Iasi	OQ601523	MH729811 (87%) CN
StlF-1	Tulcea	OQ601529	KC294199 (89%) DK
StlF-5	Tulcea	OQ601530	
StlF-7	Tulcea	OQ601531	
Ssv4-1	Suceava	OQ601526	ON644869 (89%) NL OM037396 (89%) CN
Ssv4-2	Suceava	OQ601527	
Ssv4-8	Suceava	OQ601528	
SbcS-1	Bacau	OQ601524	KX774658 (87%) DE
SbcS-6	Bacau	OQ601525	

Country abbreviations: CN, China; DE, Germany; DK, Denmark; NL, the Netherlands.

## Data Availability

The sequences are submitted in the GenBank database under accession numbers: OQ601523–OQ601531.

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
