# Peer review of "First Detection of Hepatitis E Virus (Rocahepevirus ratti Genotype C1) in Synanthropic Norway Rats (Rattus norvegicus) in Romania"

_viruses, 2023, doi:10.3390/v15061337_

Round 1

Reviewer 1 Report

viruses-2401826

First detection of Hepatitis E virus (Rocahepevirus) in wild Norway rats (Rattus norvegicus) in Romania

This study reports the first detection of Hepatitis E virus (HEV) in wild Norway rats (Rattus norvegicus) in Romania, where HEV-3 has been previously reported in pigs, wild boars, and humans. Using molecular methods, the presence of HEV RNA was investigated in liver samples collected from wild rats and other species of wild animals, with nine samples from the 52 rats tested positive for rat HEV RNA. The findings highlight the potential role of rats as a reservoir for HEV and the need to extend diagnosis of Rocahepevirus in humans with suspicion of hepatitis.

Please ensure consistency in the usage of italics for taxonomic names throughout the text, including genera, species, etc.

Figure 1 – Please mention the software program used to create the figure.

Figure 2- Please provide the bootstrap value of the phylogenetic tree and explain the significance of the (0.050) value in the figure caption.

Please include the sequence of all the primers used for the experiments in the methodology section.

It would be advisable to include the scientific name of the pigs for consistency.

In lines 135-141, The authors refer to RNA extraction, sample processing, and RT-PCR, as a previous described methodology. However, it would be helpful to provide a brief description of the protocol, as these are critical steps to ensure proper sample processing.

Lines 142, 143, and 145, The authors extensively discuss the species of HEV in the introduction, it would be important to include information about the genome composition, as it is crucial for the PCR. Please specify which fragments are amplified and provide details about how conserved they are.

Line 173 It would be beneficial to include a gel image showing the PCR products, positive and negative controls, as well as a set of representative samples.

In line 221, It would be beneficial to delve deeper into the Denmark strain and even hypothesize about the reasons for its presence in that clade or its close relationship to the Iasi strain. Please elaborate in the discussion section.

Reviewer 2 Report

The manuscript entitled “First detection of Hepatitis E virus (Rocahepevirus) in wild Norway rats (Rattus norvegicus) in Romania” by Porea et al. addresses a topic of considerable interest and topicality. It is a well-structured and well-written manuscript with attention to detail in each of its sections.

The discussion of the results obtained is articulated and supported by an adequate and recent bibliography.

I only have some very minor comments, and these are entirely up to the Authors if they wish to take them on board or not.

L.2. Considering that the rats were caught near farms and in urban or rural areas, perhaps it would be preferable to replace “wild” with “synanthropic”.

L.2. In the title and in many parts of the text Rattus norvegicus is referred to as “Norway rats” or “Norvegian rats” or “Norway ratus”. I suggest standardising the definition.

L. 24. I suggest changing “...distributed in domestic pigs” to “distributed in domestic and feral pigs”.

L. 24, L. 95, L. 105 and L. 270. Eastern/eastern

L. 28. I suggest listing which other wild animals have been investigated.

L. 121. If possible, provide some additional information on the two farms (e.g. type, indoor/outdoor, number of animals) and on the faecal pools (age of animals, number of individual samples in each pool).

Fig.1 and Table 1. Please change “Sus domesticus” with “Sus scropha” or “Sus scropha domesticus”.

L. 227. I suggest adding a reference after ''…. pigs, wild boars, and humans”.

L. 240. Please change “De Sabo et al.” with “De Sabato et al.”

L. 249. Please change “Rat-HEV” with “rat HEV”

L. 291. Please add “virus” after “hepatitis E”.

L. 292. HCV/HEV?

L. 276-286. I think that a possible cause for HEV RNA negativity is the low number of samples tested. If the authors agree with my observation, I suggest that this aspect should be discussed. Again, just a suggestion.

Overall, a very nice and interesting manuscript which I thoroughly enjoyed reading. My thanks, and best wishes for the future.

Reviewer 3 Report

This short study describes the detection and characterization of rat HEV in Romania. Recently, rat HEV has drawn increasing attention since its zoonotic transmission to humans has been confirmed in several independent studies worldwide; therefore, the topic of the study is important. However, this study provided relatively limited data on the HEV research field.  Overall, the design of the study is well-performed, and the manuscript is concise. However, there are some aspects regarding the current knowledge of rat HEV infection that needs to be updated and revised. I have the following comments for the authors’ consideration.

Major comments:

1.     Title: to be more prudent and precise, the reviewer suggests adding “genotype C1” to “hepatitis E virus”.

2.     Abstract, lines 22-24: the authors stated that rats carry both rat HEV-C1 and human/swine HEV-3, but currently, very seldom evidence support HEV-3 can infect rats. To the best of my knowledge, it preferably regards as spillover, as also mentioned by the author in maintext, unless human/swine HEV-3 is detected in the blood or tissue samples of rats. Therefore, this concept should be given more cautiousness. 

3.     Abstract, line 32: This sentence is abrupt. Consider adding “Since the rat HEV has been reported to cause zoonotic infections in humans in multiple countries,” to “it supports…”.

4.     Introduction, paragraph 2: urgently, it is noticeable that HEV-C3 derived from field mice has also been proposed in the genus Rocahepevirus. Moreover, in the last sentence, the authors have mistaken that the R. eothenomi has been detected in vole (Microtus arvalis) in Hungary and Germany. Actually, the R. eothenomi species also comprises HEV-C variants from multiple vole species (e.g., Eothenomys melanogaster and Eothenomys inez) in China (PMIDs: 30285857 and 29500690).

5.     Introduction, paragraphs 3-5: paragraphs 3 and 5 describe the presence of HEV-3 in rats and the role of rats in the transmission of HEV-3, while paragraph 4 describes the zoonotic rat HEV infection in humans. It is thus better to move paragraph 5 ahead of 4.

6.     Results: the primary limitation of the present study is there are only very partial HEV RdRp fragments (~330 bp) obtained. One or two representative complete genomes of rat HEV from Romania would be beneficial to conduct a comparative analysis with other worldwide rat HEVs in more detail. Notably, the rat HEV partial RdRp sequences from this study shared only 85-89% identities with known rat HEV strains, implying they are actually divergent.

7.     Discussion: the last paragraph seems self-contradictory to me. In the Methods section, it has been stated that the real-time RT-PCR is sensitive and nested RT-PCR is broad-spectrum for HEV detection; therefore, the methods should be good enough to detect HEV variants in other animal species besides rats. Consider rephrasing the context.

8.     There are numerous typos and spelling errors in the manuscript, which should be avoided by carefully proofreading. Additionally, please double-check that each reference is appropriately cited.

Minor comments:

1.       Abstract, line 24: “wild and domestic”

2.       Abstract, line 29: change “form” to “for”

3.       Abstract, line 29: specify the “high identity”

4.       Introduction, line 43: “Orthohepevirus A species”

5.       Introduction, lines 49 and 51: the citation of references 3 and 4 are irrelevant to the two sentences.

6.       Introduction, line 53: change “subfamilies” to “subfamily”

7.       Introduction, line 56: “Parahepevirinae

8.       Introduction, line 58: “Orthohepevirus C species”

9.       Introduction, line 58: the genotype HEV-C3 has been proposed in the Rocahepevirus genus, which should be introduced as well.

10.    Introduction, line 62: the reviewer suggests referring to the review paper (PMID: 32106525) that comprehensively discussed the host range of HEV-C1 and -C2.

11.    Introduction, line 69: the reviewer disagrees that the reference 4 and 10 support the statement.

12.    Introduction, lines 76 and 77: change “ratus” to “rats”

13.    Introduction, line 79: please double-check that human/swine HEV-3 has genuinely been detected in rats in each and every cited reference (12, 14, 15, 17-20).

14.    Introduction, line 98: “circulating HEV strains”

15.    Figure 1: please specify how the figure is generated in the legend.

16.    Materials and Methods, line 145: indicate the specific location of the amplified fragment compared with the reference HEV strain.

17.    Materials and Methods, line 165: change “strand” to “direction” 

18.    Materials and Methods, line 170: “Orthohepevirinae

19.    Results, line 189: “rat HEV-C1”

20.    Results, line 194: change “similarity” to “identity”

21.    Figure 2: there is however no information about the GI to GIII in the maintext. Consider omitting if unnecessary.

22.    Discussion, line 229: “HEV-C1”

23.    Discussion, line 237: “an”

24.    Discussion, line 252: change “similarity” to “identity”

25.    Discussion, lines 279 and 281: ensure the references 43 and 14 are correctly cited in corresponding places.

26.    Data Availability Statement: the reviewer checked that the sequences have not yet been released.

There are numerous typos and spelling errors in the manuscript, which should be avoided by carefully proofreading.

Round 2

Reviewer 3 Report

The authors have adequately addressed my concerns and queries in their revised manuscript; therefore, I have no more comments.